# Hopping and Flipping of RNA Polymerase on DNA during Recycling for Reinitiation after Intrinsic Termination in Bacterial Transcription

**DOI:** 10.3390/ijms22052398

**Published:** 2021-02-27

**Authors:** Wooyoung Kang, Seungha Hwang, Jin Young Kang, Changwon Kang, Sungchul Hohng

**Affiliations:** 1Department of Physics and Astronomy, and Institute of Applied Physics, Seoul National University, Seoul 08826, Korea; crn1860@gmail.com; 2Department of Chemistry, Korea Advanced Institute of Science and Technology, Daejeon 34141, Korea; tmdgk.hwang@kaist.ac.kr (S.H.); jykang59@kaist.ac.kr (J.Y.K.); 3Department of Biological Sciences, Korea Advanced Institute of Science and Technology, Daejeon 34141, Korea

**Keywords:** Cy3, Cy5, *Escherichia coli*, fluorescent labeling, fluorescent transcription complex, one-dimensional facilitated diffusion, protein-induced fluorescence enhancement, single-molecule fluorescence, transcription termination

## Abstract

Two different molecular mechanisms, sliding and hopping, are employed by DNA-binding proteins for their one-dimensional facilitated diffusion on nonspecific DNA regions until reaching their specific target sequences. While it has been controversial whether RNA polymerases (RNAPs) use one-dimensional diffusion in targeting their promoters for transcription initiation, two recent single-molecule studies discovered that post-terminational RNAPs use one-dimensional diffusion for their reinitiation on the same DNA molecules. *Escherichia coli* RNAP, after synthesizing and releasing product RNA at intrinsic termination, mostly remains bound on DNA and diffuses in both forward and backward directions for recycling, which facilitates reinitiation on nearby promoters. However, it has remained unsolved which mechanism of one-dimensional diffusion is employed by recycling RNAP between termination and reinitiation. Single-molecule fluorescence measurements in this study reveal that post-terminational RNAPs undergo hopping diffusion during recycling on DNA, as their one-dimensional diffusion coefficients increase with rising salt concentrations. We additionally find that reinitiation can occur on promoters positioned in sense and antisense orientations with comparable efficiencies, so reinitiation efficiency depends primarily on distance rather than direction of recycling diffusion. This additional finding confirms that orientation change or flipping of RNAP with respect to DNA efficiently occurs as expected from hopping diffusion.

## 1. Introduction

To properly function in time, sequence-specific DNA-binding proteins need to find their target sequences rapidly with high affinity and specificity. Target binding of *Escherichia coli* lac repressor was experimentally observed to be much faster than the association rates expected from three-dimensional (3D) diffusion-limited processes [1], and was then proposed to involve one-dimensional (1D) facilitated diffusion of the protein on DNA [2]. The 1D diffusion of proteins on DNA includes two distinct mechanisms, hopping and sliding.

In the hopping mechanism, proteins transiently dissociate from DNAs for locally limited 3D diffusion and shortly bind back to DNA at nearby segments, and their microscopic dissociation and reassociation are repeated. Hopping does not necessarily follow the helical backbone path of DNA. To compensate for the dissociation of hopping proteins, counter-ions bind to DNAs. Because the binding equilibrium is shifted to protein-dissociated state under high salt conditions, 1D diffusion coefficients of any hopping proteins increase with rising salt concentrations [3,4,5,6].

In the sliding mechanism, by contrast, proteins move along DNAs without any detachment. Sliding in principle follows the DNA helical path but repetitively replaces local contacts between proteins and DNA segments. Without any transient microscopic dissociation of proteins from DNAs, 1D diffusion coefficients of any sliding proteins are little affected by salt concentrations [7,8,9,10].

It has been controversial whether RNA polymerase (RNAP) uses 1D facilitated diffusion in promoter search for transcription initiation or not. Earlier works using gel electrophoresis, scanning force microscopy, or single-molecule visualization contended that 1D diffusion facilitates the pre-initiation promoter search of RNAPs [11,12,13,14]. On the other hand, other studies using single-molecule fluorescence imaging argued that DNA binding duration of RNAP is too short for 1D facilitated diffusion to be efficient, and that pre-initiation promoter search of RNAP is dominated by 3D diffusion [15,16].

Recently, on the other hand, two independent single-molecule fluorescence studies concurrently discovered 1D diffusion of post-terminational RNAP in *E. coli* transcription system [17,18]. After releasing RNA at intrinsic termination, RNAP mostly remains on DNA and one-dimensionally diffuses in both downstream and upstream directions, and recycles for reinitiation at nearby promoters. Recycling RNAP refers to the RNAP that has synthesized and released RNA (post-terminational), but remains DNA-bound and diffuses on DNA rather than falling off DNA. Reinitiation refers to the initiation performed by recycling RNAP reaching a promoter via 1D diffusion rather than by fresh or free RNAP associating with a promoter via 3D diffusion.

In the two studies, however, it was not resolved which 1D diffusion mechanism, sliding or hopping, is used by recycling RNAP between termination and reinitiation. This question is intriguing, particularly because the two observations on 1D diffusion of recycling RNAP are seemingly consistent with different mechanisms of diffusion, one observation with sliding and the other with hopping. One observation was that 1D diffusion of recycling RNAP is blocked by a roadblock bound on DNA [17]. This alludes the sliding mechanism, where RNAP does not leave DNA microscopically or temporarily, because the roadblock could interfere with sliding of protein along DNA [9] more efficiently than hopping [3,5,6].

The other observation was that transcription direction can be reversed by recycling RNAP for reinitiation from an antisense-oriented promoter on the same template DNA molecule [18]. This observation implies the hopping mechanism, where repetitive microscopic dissociation of protein from DNA can allow for RNAP or a local DNA segment to freely rotate or flip on each other, because hopping does not always keep close contacts between proteins and DNAs, and thereby could allow for flipping of proteins with respect to DNAs more efficiently than sliding.

Here we report multiple single-molecule fluorescence experiments of *E. coli* RNAP transcription performed under various salt conditions. The 1D diffusion coefficient of recycling RNAP increases with rising salt concentrations, demonstrating that recycling RNAP undergoes hopping diffusion. We additionally find that reinitiation efficiencies are similar when recycling RNAP reinitiates transcription from two oppositely oriented promoters on a DNA template, supporting that recycling RNAP can flip as efficiently as expected from hopping diffusion mechanism.

## 2. Results

### 2.1. Salt Dependent Retention of Recycling RNAP on DNA

To determine the 1D diffusion coefficient of recycling RNAP on DNA, we first measured its DNA-bound durations under various salt conditions (Materials and Methods) as its experimental scheme is depicted in Appendix A. We prepared stalled transcription complexes by incubating a specially designed template DNA, ATP, CTP, GTP, Cy3-labeled ApU, Cy5-labeled *E. coli* RNAP core enzyme, and unlabeled sigma factor (σ^70^) at 37 °C for 30 min in a transcription reaction buffer. Transcription is initiated preferentially with Cy3-ApU, which is incorporated into the +1 and +2 positions of RNA, so transcript RNA is nascently labeled with Cy3 at the 5’ end. Because UTP is not included in the initiation reaction buffer, transcription is stalled in complexes with 12-nucleotide long RNA waiting for the missing UTP.

The template DNA of 2424 bp comprises a 50-bp upstream part including the strong promoter A1 of bacteriophage T7, a 38-bp transcription unit with the intrinsic terminator tR2 of phage λ, and a 2336-bp downstream part (Figure 1a). The major termination site (TS) is the +38 position, as the experimental scheme is basically the same as our previous experiments [17], except that the downstream part of TS is much longer this time in order to minimize the effect of RNAP adhering to DNA ends on the retention of post-terminational RNAP on DNA, and that Cy5 labeling is on RNAP rather than DNA this time in order to monitor the dissociation of recycling RNAP from anywhere along DNA rather than its association with the dye-labeled end position of DNA.

The DNA upstream end, i.e., the 5’ end of nontemplate (sense) strand, is labeled with biotin for surface immobilization, and the downstream end, i.e., the 5’ end of template (antisense) strand, is free. As depicted in Appendix A, the stalled complexes with 12-mer RNA were immobilized on polymer-coated quartz slides using biotin-streptavidin conjugation, and extensively washed for removal of unimmobilized complexes and unbound reaction components. Transcription elongation was resumed by injection of a reaction buffer containing all four ribonucleotides (NTPs), while fluorescence images from individual spots of immobilized complexes were taken using a total-internal-reflection fluorescence microscope. A schematic diagram of the microscope is shown in Appendix A.

The elongation reaction buffers each contained 2 mM MgCl_2_ and a varying concentration of NaCl. Representative time traces of Cy3-RNA (green) and Cy5-RNAP (red) fluorescence at Cy3 excitation (top) and Cy5 excitation (bottom) from experiments with 150 mM NaCl are shown in Figure 1b. Shortly after NTP addition, the Cy3 signal disappears from some spots of immobilized complexes, indicating that transcript RNAs are released from them at termination of transcription.

The Cy5 signal disappears from spots when RNAP is dissociated from immobilized unlabeled DNA. While Cy3 and Cy5 signals vanish concurrently in some complexes of termination, Cy5 vanishes sometime after Cy3 vanishes in most complexes. During the period (*t*_retention_) between Cy3 and Cy5 vanishing, post-terminational RNAP retains on DNA, so *t*_retention_ is DNA-bound duration or DNA retention time of recycling RNAP after termination. As expected with any DNA binding proteins, *t*_retention_ of recycling RNAP decreases with rising NaCl concentrations (Figure 1c, Appendix A), i.e., recycling RNAP gets off DNA sooner at higher salt concentrations. Since the photobleaching time of Cy5 (750 s) is much longer than *t*_retention_ (Appendix A), retention time measurements are little affected by photobleaching.

### 2.2. Salt Dependent 1D Diffusion Coefficient of Recycling RNAP

To study whether or not salt concentrations affect the 1D diffusion coefficient of recycling RNAP using single-molecule fluorescence imaging, we used the fluorescent elongation complexes that we previously constructed [17]. Cy3 labeling is still on nascent RNA at the 5’ end, but Cy5 labeling is not on RNAP but on immobilized DNA at the downstream end, while biotin labeling of DNA is still on the upstream end (Figure 2a). Six different DNA templates each have a TS-downstream part of varying length, 15, 62, 112, 212, 312, or 512 bp, and are denoted by L+# with TS-downstream base-pair numbers.

Elongation of the initially stalled complexes with 12-mer RNA was resumed by injection of an elongation reaction buffer containing all four NTPs, 2 mM MgCl_2_ and a varying concentration of NaCl, as the experimental scheme is shown in Appendix A. Representative fluorescence time traces from the experiments with L+212 with 212-bp TS-downstream part at 150 mM NaCl are presented in Figure 2b–d. These data are generally consistent with the data of our previous experiments using L+15 instead of L+212 under various salt conditions or using DNAs with various TS-downstream lengths under a salt condition [17].

Figure 2b shows TS-readthrough cases (61%, 150/246), in which protein-induced fluorescence enhancement (PIFE) of Cy5 is observed without Cy3-RNA release. Cy5 PIFE is caused by RNAPs contacting the Cy5 that is labeled at the downstream end of DNA. Thus, elongating RNAPs ignore TS and extend the transcription unit to reach the Cy5-labeled downstream end without releasing extended RNAs. On the other hand, TS-termination cases, where Cy3-RNA signal disappears shortly after NTP injection, exhibit two different types of traces after termination (Figure 2c,d).

In type I post-termination cases (Figure 2c, 32%, 79/246), Cy5 PIFE starts sometime after Cy3-RNA release, being delayed by *t*_delay_. This Cy5 PIFE indicates that RNA-free recycling RNAPs reach the Cy5-labeled downstream end of DNA via 1D diffusion rather than 3D diffusion as previously described [17]. Thus, PIFE delay time *t*_delay_ is the time taken by recycling RNAPs to diffuse along DNA from TS, where Cy3 vanishes, to the downstream end, where Cy5 PIFE occurs. PIFE delay tends to decrease with rising salt concentrations (Appendix A). In type II post-termination cases (Figure 2d, 7%, 17/246), no PIFE is observed after RNA release, indicating that some post-terminational RNAPs get off DNA before reaching the Cy5-labeled end.

To measure 1D diffusion coefficients of recycling RNAPs, type I post-termination cases were compared among six different templates with varying length, L+15 through L+512. With increasing TS-downstream lengths (i.e., distances from TS to the Cy5-labeled downstream end), PIFE occurs in fewer complexes (decreasing occurrence in Figure 2e) and starts later (increasing delay in Figure 2f) as previously described [17]. The 1D diffusion coefficient was estimated in a 1D random walk analysis using the correlation between TS-downstream length and PIFE occurrence or delay (Materials and Methods). For example, from 150 mM NaCl experiments with varying length DNA templates, two estimates for 1D diffusion coefficient of recycling RNAP using PIFE occurrence and delay data are similar to each other, (6.4 ± 1.4) × 10^−4^ and (6.9 ± 2.0) × 10^−4^ μm^2^/s, respectively.

To examine salt concentration dependency of the 1D diffusion coefficient, we repeated the set of 150 mM NaCl experiments with DNA length variation separately at varying concentrations of NaCl, 20, 50, 100, and 200 mM (Appendix A). The 1D diffusion coefficients of recycling RNAPs that were estimated using PIFE occurrence data increase almost linearly with rising concentrations of NaCl from 20 to 200 mM (red line in Figure 2g), demonstrating that recycling RNAPs adopt hopping mechanism in post-terminational diffusion on DNA. These data, however, do not rule out possible existence of intermittent sliding simply because sliding in principle is little affected by salt concentrations, and suggest that 1D diffusion of recycling RNAP is entirely or partly hopping rather than entirely sliding.

The diffusion coefficients estimated using PIFE delay data (green line in Figure 2g) additionally support hopping mechanism, although less evidently than PIFE occurrence data apparently due to broader error ranges of PIFE delay data. At every NaCl concentration used in the experiments of this study, the error range is broader for the coefficient estimates using PIFE delay data than for those using PIFE occurrence data (Figure 2g). While statistical analyses with fewer samples tend to yield greater margins of sampling error, PIFE timing complexes are inevitably fewer than PIFE occurrence counting complexes. It is because termination cases without Cy5 PIFE (some recycling RNAPs get off DNA before reaching Cy5 at an end) are excluded in PIFE timing measurements but included in the denominator counts of PIFE occurrences.

### 2.3. Sense and Antisense Reinitiation Efficiencies of Recycling RNAP

Recycling RNAP has been demonstrated to diffuse on DNA in both downward (forward in the transcription direction) and upward (backward) directions presumably in a random fashion (Appendix A) until it reaches either downstream or upstream end of DNA, where it is trapped for a prolonged time. A 1D diffusion-blocking roadblock placed at downstream of TS efficiently hampers recycling RNAP from reaching the downstream end but not reaching the upstream end [17].

Furthermore, recycling RNAPs mostly possess σ factor, and even those having lost it can uptake it again while diffusing on DNA [17]. Recycling RNAP is thereby capable of reinitiating transcription on the original promoter located upstream of TS (upstream reinitiation) [17], on a TS-downstream promoter placed in the same transcription orientation (downstream reinitiation) [17], or on a TS-upstream promoter positioned in the opposite orientation (antisense reinitiation) [18], while the reinitiation efficiency is enhanced by supplement of σ factor [17].

As 1D diffusion of recycling RNAP turns out in this study to involve hopping, which in principle can allow for orientation change or flipping of RNAP with respect to DNA, we next characterized how efficiently recycling RNAP can flip on immobilized DNA during post-terminational hopping for reinitiation. The flipping efficiency was indirectly measured in this study by quantitatively comparing reinitiation efficiencies of two oppositely oriented promoters placed on the same DNA molecule. In addition to the original 38-bp transcription unit under promoter 1, L+2P template was constructed to contain another transcription unit under promoter 2 at downstream of TS for antisense-oriented transcription and a probing part with five repeats of a 21-bp sequence between the two promoters, and was labeled with biotin at the upstream end (Figure 3a).

With L+2P template, the first-round transcripts initiating at the +1 position under promoter 1 are nascently labeled with Cy3-ApU, which is used for monitoring termination and readthrough occurring at TS in the first-round transcription. TS-readthrough RNAPs likely proceed to the downstream end while transcribing the sense-transcript probing part (from +59 to +162), but TS-termination RNAPs recycle on DNA diffusing in both directions unless fall off DNA. The RNA-free recycling RNAPs can undergo sense reinitiation at the TS-upstream promoter 1 (downward from +1, transcribing the sense probing part) or antisense reinitiation at the TS-downstream promoter 2 (upward from +164, transcribing the antisense probing part from +162 to +59), unless fall off DNA. Thus, termination at TS, recycling, and reinitiation at promoter 1 can form a cycle, while readthrough at TS, reinitiation at promoter 2 or exit from recycling breaks the cycle (Appendix A).

Sense and antisense reinitiation events after termination at TS can be measured distinguishably by separately monitoring the binding of two different Cy5-labeled single-strand probes to sense and antisense transcripts, respectively (Materials and Methods, Appendix A). Here, Cy5 labeling is on the probes rather than DNA or RNAP. Although multiple rounds of reinitiation can occur at promoter 1 (Appendix A), only the very first-round transcript RNA is labeled with Cy3-ApU, because the dye is washed away after the initial stalling (Appendix A).

When the reinitiation experiments were performed with supplement of σ^70^ (3 μM) as previously described [17] but with L+2P template in this study, three types of events were observed with the sense probe (30 nM). (1) Cy3-RNA signal disappearance is observed and followed by Cy5 sense-transcript probing signal appearance (frequency of 0.068 ± 0.016). This happens when transcription reinitiation occurs at promoter 1 after termination at TS. (2) Cy3 signal disappears without Cy5 signal appearance (0.526 ± 0.028). This is observed when no reinitiation occurs due to RNAP being dissociated off DNA or inactivated on DNA upon or after termination at TS. (3) Cy5 signal appears without Cy3 signal disappearance indicating transcription readthrough at TS (0.406 ± 0.027).

With the antisense probe (30 nM) in separate experiments, only type 1 (0.099 ± 0.032) and type 2 (0.901 ± 0.032) events are observed depending on whether reinitiation occurs on promoter 2 or not. Because the antisense probe does not bind TS-readthrough sense transcripts, type 3 is not observed with it.

Transcription reinitiation efficiency should not be estimated simply as the ratio of the type 1 frequency to the sum of type 1 and type 2 frequencies, because probe binding is not complete, and because multiple rounds of termination and reinitiation can occur (Appendix A). However, we took all these complications into estimation of actual reinitiation efficiencies (Materials and Methods) as previously described [17].

Reinitiation efficiency is estimated to be 46.8 ± 4.1% for sense-oriented promoter 1 using the sense-transcript probing data and 24.6 ± 8.0% for antisense-oriented promoter 2 using the antisense probe data. Recycling RNAP would recognize farther promoters a priori less efficiently, as it is demonstrated to reach farther end of DNA less efficiently in Figure 2e. As transcription start site (+164) of the antisense promoter 2 is positioned farther away from TS (+38) than that (+1) of the sense promoter 1 on L+2P template, 126 bp versus 37 bp away, we can approximate that distance-independent efficiencies of reinitiation are comparable for sense- and antisense-oriented promoters on the same DNA molecule. These data suggest that flipping of recycling RNAPs or local DNA segments is little impeded during the post-terminational 1D diffusion, and that reinitiation efficiency would depend more on distance of recycling diffusion than orientation of RNAP on DNA.

## 3. Discussion

In this study, we discover that hopping mechanism of molecular diffusion on DNA is utilized by recycling RNAP during its 1D facilitated diffusion after intrinsic termination in bacterial transcription. This finding is based on that 1D diffusion coefficient of post-terminational recycling RNAP increases in proportion to salt concentration, when the diffusion coefficients of *E. coli* RNAP were measured in multiple sets of single-molecule experiments with various distances of diffusion on DNA at various concentrations of NaCl in diffusion space.

Furthermore, we additionally find that similar efficiencies are achieved by recycling RNAP for reinitiation on sense- and antisense-oriented promoters on the same DNA molecule. As reinitiation on antisense promoters requires flipping of RNAPs or local DNA segments with respect to each other, this additional finding supports hopping mechanism because flipping is hardly facilitated by sliding, which is the other mechanism of molecular diffusion on DNA. This finding additionally suggests that reinitiation efficiency would depend much less on direction than distance of recycling diffusion, because reinitiation with or without requiring the flip is comparable in its efficiency.

All of the findings of this study together strongly support hopping of recycling RNAPs, but do not reject the possibility that hopping is interspersed with intermittent sliding throughout recycling diffusion. It is because the presence of sliding, which is little affected by salt concentrations, can hardly be detected in salt dependency of 1D diffusion coefficient, unless the diffusion is solely sliding. Without evidence, however, we think it is even likely that recycling RNAP not only hops but also slides as some other proteins do both [19], because sliding would be a priori more effective than hopping for sequence-specific interaction of RNAPs with cognate promoters for reinitiation.

It would be sensible that hopping and sliding are utilized in optimally balanced portions by recycling RNAPs for reinitiation. Hopping could be faster than sliding in reaching promoters, sliding would be more accurate than hopping in recognizing cognate sequences, and flipping would be required for antisense reinitiation. Thus, reinitiation efficiency, which depends on distance between TS of one transcription unit and transcription start site of another unit, would be optimized for distantly positioned or oppositely oriented units respectively with more hopping or flipping than for adjacent units in the same orientation. Their balanced portions need to be determined, while it is curious whether the portions are subject to regulation as reinitiation efficiency would depend on spatial arrangement of adjacent transcription units. Their relative frequencies in a round of recycling diffusion can be measured by single-molecule fluorescence experiments [19], but requiring different settings from this study.

As it is evidenced in this study that flipping of recycling RNAP is facilitated by hopping mechanism, reinitiation of oppositely oriented nearby transcription units would produce antisense transcripts. Many antisense transcription units are found in bacteria, archaea, and eukaryotes [20], although recycling and reinitiation of transcription have not been verified to occur in vivo or in archaea and eukaryotes yet. In the *E. coli* genome, about 30% of transcription units are antisense to other overlapping or nearby genes or operons [21]. In the human genome, many long noncoding RNA genes are antisense to one or two overlapping or nearby protein-coding genes. It remains to be examined whether these antisense-oriented transcription units are expressed in vivo by recycling RNAP that is flipped for reinitiation after termination of nearby transcription units.

Moreover, antisense transcription can participate in gene expression regulation as well. It can work by several mechanisms, although its role is still elusive. Previous studies have shown that antisense transcription relates to thresholds of gene expression switches [22,23,24]. Flipping in recycling diffusion of RNAP along DNA can allow antisense transcription to occur immediately after sense transcription without RNAP dissociation. Thus, this coupling could facilitate gene regulation by antisense transcription.

In summary, hopping mechanism is evidently utilized by recycling RNAP, which has synthesized and released product transcript RNA from a bacterial transcription unit with an intrinsic terminator, during its 1D facilitated bidirectional diffusion for reinitiation from nearby promoters on the same DNA molecule, probably in addition to sliding mechanism presumably used for final specific interactions with cognate promoter sequences. Furthermore, hopping mechanism efficiently facilitates flipping of RNAPs or local DNA segments on each other to render similar efficiencies in sense and antisense reinitiation, so reinitiation efficiency depends much more on distance than direction of hopping diffusion.

## 4. Materials and Methods

### 4.1. Single-Molecule Experiments of Transcription Termination

Stalled transcription complexes were constructed by incubation of a DNA template (50 nM) with Cy3-labeled ApU (250 μM, TriLink BioTechnologies, San Diego, CA, USA), ATP, CTP, GTP (20 μM each, GE Healthcare, Seoul, Korea), and unlabeled *E. coli* RNAP holoenzyme (20 nM) purchased from New England Biolabs (NEB), Ipswich, MA, USA in a buffer consisting of 20 mM NaCl, 20 mM MgCl_2_, 10 mM Tris-HCl, pH 8.0, and 1 mM dithiothreitol at 37 °C for 30 min. For RNAP retention measurements, custom-purified Cy5-labeled RNAP core enzyme (40 nM) and unlabeled σ^70^ (1 μM) were used instead of the unlabeled holoenzyme.

Using piranha solution (a 2:1 mixture of sulfuric acid and hydrogen peroxide), excess organic residues were removed off quartz slides, which were then soaked in 3-aminopropyl trimethoxysilane (United Chemical Technologies, Bristol, PA, USA). Next, we incubated the slides in a 1:40 mixture of biotin-PEG-5000 and m-PEG-5000 (Laysan Bio, Arab, AL, USA), and treated them with 0.2 mg/mL streptavidin (Invitrogen, Thermo Fisher Scientific, Waltham, MA, USA) for 5 min, so that the stalled transcription complexes could be immobilized on them via biotin-streptavidin conjugation.

We used a homemade total-internal-reflection fluorescence microscope equipped with a 532-nm green laser (EXLSR-532-50-CDRH, Spectra-Physics, MKS Instruments, Andover, MA, USA) for Cy3 excitation and a 640-nm red laser (EXLSR-640C-60, Spectra-Physics) for Cy5 excitation (Appendix A). As an imaging device, an electron-multiplying charge-coupled camera (Ixon DV897, Andor Technology, Oxford Instruments, Belfast, United Kingdom) was used. In all experiments performed at 37 °C, the exposure time was 0.05 s for PIFE measurements, 0.1 s for RNAP retention measurements, and 0.2 s for reinitiation experiments. The time resolution was 0.1 s, 0.2 s, and 0.4 s, respectively, because we used an alternating laser excitation system in order to clearly distinguish Cy5 PIFE from fluorescence resonance energy transfer. All equipment was controlled by a customized C# program.

Transcription elongation was resumed with 200 μM NTP each in an oxygen scavenging buffer (a varying concentration of NaCl, 2 mM MgCl_2_, 10 mM Tris-HCl, pH 8.0, 1 mM dithiothreitol, 5 mM 3,4-protocatechuic acid, and 100 nM protocatechuate-3,4-diocygenase). Before resuming elongation, we extensively washed the complexes with the oxygen scavenging buffer without any NTP to remove all the components that did not constitute elongation complexes and to change the buffer. For reinitiation experiments (Appendix A), we used a Cy5-labeled sense probe (30 nM, 5′-Cy5-TGTGT GTGGT CTGTG GTGTC T-3′) or an antisense probe (30 nM, 5′-Cy5-AGACA CCACA GACCA CACAC A-3′) in the oxygen scavenging buffer with 150 mM NaCl and 3 μM σ^70^ factor.

### 4.2. A 1D Diffusion Model for RNAP Recycling on DNA after Termination

RNAP movement on DNA is modeled as a 1D random walk on linear DNA with only the upstream (left-side) end being immobilized (i.e., reflecting end) as described in Appendix A. The probability *P*(*x*) that RNAP starting a random walk from any position *x* reaches the downstream (right-side) end (i.e., absorbing end) satisfies Equation (1).
(1)$$P\left(x\right)=P_{R}*P\left(x+\delta \right)+P_{L}*P\left(x-\delta \right)$$
where *P*_R_, and *P*_L_ are the probabilities that RNAP at any position along DNA moves to the right and left sides, respectively. Equation (1) can be rewritten as Equation (2).
(2)$$\frac{2\tau }{\delta^{2}t_{\mathrm{retention}}}P\left(x\right)=\left(1-\frac{\tau }{t_{\mathrm{retention}}}\right)\left[\frac{\left(P\left(x+\delta \right)-P\left(x\right)\right)-\left(P\left(x\right)-P\left(x-\delta \right)\right)}{2\delta^{2}}\right]$$

In the limit that *δ* approaches to 0, Equation (2) becomes Equation (3).
(3)$$P\left(x\right)=t_{\mathrm{retention}}D\frac{\partial^{2}}{\partial x^{2}}P\left(x\right)$$
where *D* is a 1D diffusion coefficient defined as *δ*^2^/2*τ*.

If RNAP starts random walk motion at the absorbing end (*x* = *x*_R_), the RNAP that has not dissociated from DNA upon termination reaches the same absorbing end with 100% probability. If RNAP starts random walk motion at the reflecting end (*x* = *x*_L_), the derivative of *P*(*x*) with respect to *x* should be 0 from Equation (1). Therefore, we obtain the following two boundary conditions.
(4)$$P\left(x_{R}\right)=\varepsilon_{R}$$
(5)$${\frac{dP\left(x\right)}{dx}}_{x=x_{L}}=0$$
where *ε*_R_ is the probability of RNAP retention on DNA among all TS-termination cases and can be referred to as recycling efficiency (Appendix A). With these boundary conditions, the partial differential Equation (3) can be solved as Equation (6).
(6)$$P\left(x\right)=\varepsilon_{R}\frac{\cosh\left(x-x_{L}/l_{D}\right)}{\cosh\left(\left(x_{R}-x\right)/l_{D}\right)}$$
where *l*_D_^2^ is equal to *t*_retention_ × *D*. For DNA templates used in the experiments of this study, *x*_L_ = −88, *x* = 0 at TS, and *x*_R_ = *b* in the unit of base pair, where *b* is the distance between TS and the Cy5-labeled end. Therefore, Equation (6) becomes a function of *b* as Equation (7).
(7)$$P\left(b\right)=\varepsilon_{R}\frac{\cosh\left(88/l_{D}\right)}{\cosh\left(b+88/l_{D}\right)}$$


Diffusion coefficients can be estimated by fitting PIFE occurrence data to Equation (7).

On the other hand, the mean time for RNAP to reach the absorbing end, *W*(*x*), which corresponds to the Cy5 PIFE delay time *t*_delay_ in Figure 2c, satisfies Equation (8).
(8)$$W\left(x\right)=\;\tau \;+\;P_{R}W\left(x+\delta \right)+P_{L}W\left(x-\delta \right)$$


In the limit that *δ*→0, Equation (8) becomes a partial differential as Equation (9).
(9)$$W\left(x\right)\;-t_{\mathrm{retention}}\;=\;t_{\mathrm{retention}}D\frac{\partial^{2}}{\partial x^{2}}W\left(x\right)$$


As similarly described in the above, *W*(*x*) should satisfy the following two boundary conditions.
(10)$$W\left(x_{R}\right)=0$$
(11)$${\frac{dW\left(x\right)}{dx}}_{x=x_{L}}=0$$


With these boundary conditions, Equation (9) can be solved as Equation (12).
(12)$$W\left(x\right)=t_{\mathrm{retention}}\left(1-\frac{\cosh\left(x-x_{L}/l_{D}\right)}{\cosh\left(x_{R}-x_{L}/l_{D}\right)}\right)$$


By substituting *x*_L_ = −88, *x* = 0, and *x*_R_ = *b*, Equation (12) becomes a function of *b* as Equation (13).
(13)$$W\left(b\right)=t_{\mathrm{retention}}\left(1-\frac{\cosh\left(88/l_{D}\right)}{\cosh\left(\left(88+b\right)/l_{D}\right)}\right)$$

### 4.3. Calculation of Transcription Reinitiation Efficiencies

When reinitiation occurs at promoter 1 on DNA template L+2P (Figure 3a), RNAP encounters an intrinsic terminator again (Appendix A). Since some of these RNAP can go through the next rounds of reinitiation after termination at TS, the actual reinitiation probabilities at promoter 1 (*P*_1_) and promoter 2 (*P*_2_) are not respectively equal to the measured values of sense transcription (*P*_sense_) and antisense transcription (*P*_antisense_), but they should satisfy the following two relations.
(14)$$P_{\mathrm{sense}}=\varepsilon_{\mathrm{PS}}\overset{\infty }{\underset{n=0}{\sum }}\varepsilon_{R}P_{1}\left(1-\varepsilon_{T}\right){\left(\varepsilon_{R}P_{1}\varepsilon_{T}\right)}^{n}=\varepsilon_{\mathrm{PS}}\frac{\varepsilon_{R}P_{1}\left(1-\varepsilon_{T}\right)}{1-\varepsilon_{R}P_{1}\varepsilon_{T}}$$
(15)$$P_{\mathrm{antisense}}=\varepsilon_{\mathrm{PA}}\overset{\infty }{\underset{n=0}{\sum }}\varepsilon_{R}P_{2}{\left(\varepsilon_{R}P_{1}\varepsilon_{T}\right)}^{n}=\varepsilon_{\mathrm{PA}}\frac{\varepsilon_{R}P_{2}}{1-\varepsilon_{R}P_{1}\varepsilon_{T}}$$
where *ε*_PS_ is probing efficiency of the sense probe, *ε*_PA_ is probing efficiency of the antisense probe, and *ε*_T_ is TS-termination efficiency. We can derive the following equations for *P*_1_ and *P*_2_.
(16)$$P_{1}=\frac{1}{\varepsilon_{R}}\times \frac{P_{\mathrm{sense}}}{\varepsilon_{\mathrm{PS}}\left(1-\varepsilon_{T}\right)+P_{\mathrm{sense}}\varepsilon_{T}}$$
(17)$$P_{2}=\frac{P_{\mathrm{antisense}}}{\varepsilon_{R}\varepsilon_{\mathrm{PA}}}\left(1-\varepsilon_{R}P_{1}\varepsilon_{T}\right)$$

*ε*_PS_ of the sense probe can be estimated from the experimental data obtained using the sense probe as follows. Three types of events are observed using the sense probe as shown in Figure 3, and their relative frequencies are 0.068 (type 1), 0.526 (type 2), and 0.406 (type 3). Types 1 and 2 together represent all termination events, and their sum frequency is 0.594. As the termination efficiency we measured is *ε*_T_ = 39%, the frequency of total events is 1.523 (=0.594/0.39), and the frequency of readthrough events is 0.929 (=1.523 − 0.594). Due to incompleteness of the probing, only 0.406 of all readthrough events were observed as type 3, so *ε*_PS_ is 43.7% (=0.406/0.929). Consequently, *ε*_R_ is estimated to be 78.8% by extrapolation of the curve to *x* = 0 (at TS) in Figure 2e. Then, *P*_1_ is estimated to be 46.8 ± 4.1% using the experimental values of *P*_sense_ in Equation (16).

*ε*_PA_ of the antisense probe cannot be directly measured or estimated in the experimental scheme of this study, but one can approximate that probing efficiencies are the same for the sense and antisense probes, which are complementary to each other in sequence, so ε_PA_ ≈ ε_PS_ = 43.7%. Then, *P*_2_ is approximated to be 24.6 ± 8.0% using the experimental values of *P*_antisense_ and the above-estimated values of *P*_1_ in Equation (17).

### 4.4. Cy5 Labeling of E. coli Core RNAP

*E. coli* RNAP subunits, RpoA (α), RpoB (β), RpoC (β’) with a C-terminal SNAP tag, and RpoZ (ω) were produced together in *E. coli* BL21(DE3) cell and purified as previously described [25]. Reconstituted SNAP-tagged RNAP (3.9 mg/mL) was incubated with Cy5-labeled O^6^-benzylguanine at room temperature for 2 h in a storage buffer (50 mM Tris-HCl, pH 7.5, 50 mM NaCl, and 5 mM ethylenediaminetetraacetic acid). O^6^-benzylguanine was labeled with Cy5 by incubation of BG-NH2 (NEB) with Cy5-NHS ester (12.5 μg/μL, GE Healthcare) at room temperature for 1.5 h. Unreacted molecules of Cy5-labeled O^6^-benzylguanine, BG-NH2, and Cy5-NHS ester were removed using Amicon Ultra Centrifugal Filters (Merck Korea, Seoul, Korea).

### 4.5. Preparation of Transcription Templates

Transcription template DNAs were constructed primarily by polymerase chain reactions using relevant amplification templates (Appendix A) and primers (Appendix A): L+15 was prepared using UP8_template, forward_primer_biotin, and reverse_primer_L+15; L+62 using UP8_template, forward_primer_biotin, and reverse_primer_L+62; L+112 using UP8_template_2, forward_primer_biotin, and reverse_primer_L+112. For L+212 and L+312 constructions, L+112 and additional_part_1 were annealed with DNA_splint_1 by cooling from 90 to 30 °C for 120 min in the annealing buffer (10 mM Tris-HCl, pH 8.0, and 50 mM NaCl), ligated using T4 DNA ligase 2 (NEB), and used for amplification reactions. L+212 was prepared using forward_primer_biotin and reverse_primer_L+212; L+312 using forward_primer_biotin and reverse_primer_L+312. For L+512 construction, L+312 and additional_ part_2 were annealed with DNA_splint_2 for ligation and amplified using forward_primer_biotin and reverse_primer_L+512.

For L+2P and the DNA template for the retention time measurement of recycling RNAP, long_tail template with a HindIII recognition sequence was prepared using lambda DNA (NEB), lambda_forward_primer, and lambda_reverse_primer. To construct the upstream part of L+2P template, L+2P_part1 and L+2P_part2 were annealed with L+2P_splint by cooling from 90 to 30 °C for 120 min in the annealing buffer for ligation. Long_tail DNA, the upstream part of L+2P, and L+512 were each digested with HindIII (NEB) at 37 °C for 1 h in the CutSmart™ buffer (NEB), before HindIII was inactivated at 80 °C for 20 min. Long_tail DNA were annealed with cleaved L+512 or the upstream part of L+2P and used for amplification. L+2P was prepared using forward_primer_biotin_α and lambda_reverse_primer, and the DNA template for the retention time measurement of recycling RNAP using forward_primer_biotin and lambda_forward_primer.

## Figures and Tables

**Figure 1 ijms-22-02398-f001:**
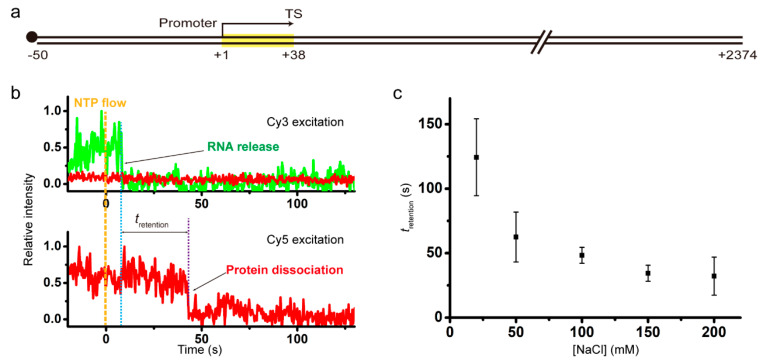
(**a**) A 2424-bp DNA template including a 38-bp transcription unit (yellow box) under T7A1 promoter, ending at the major termination site (TS) of tR2 terminator. The DNA is labeled with biotin (black dot) at the upstream end for immobilization. (**b**) Representative fluorescence traces. RNAP retention time, *t*_retention_, refers to the time difference between Cy3 vanishing due to RNA release and Cy5 vanishing due to RNAP dissociation in the termination cases with sequential dissociation of RNA and RNAP from DNA. (**c**) The RNAP retention times plotted against NaCl concentrations. After intrinsic termination, recycling RNAP falls off DNA sooner at higher NaCl concentrations.

**Figure 2 ijms-22-02398-f002:**
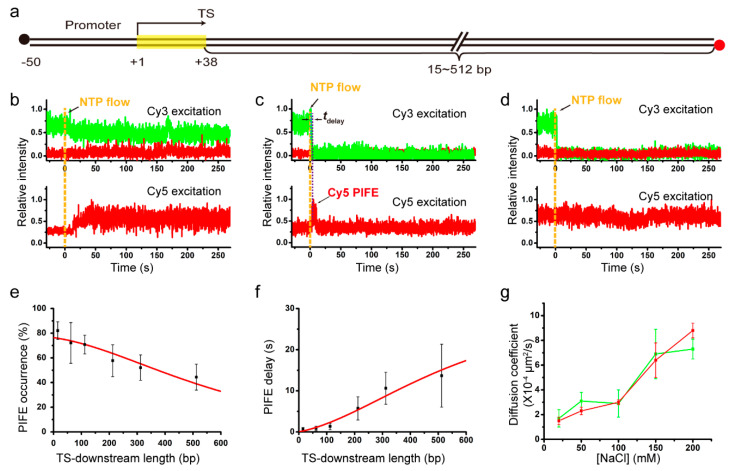
(**a**) DNA with varying length of TS-downstream part. Template DNAs were labeled with Cy5 at the downstream end and immobilized using biotin (black) at the upstream end. Transcript RNAs were nascently labeled with Cy3 at the 5’ end. (**b**–**d**) Representative fluorescence traces were taken from complexes with DNA template L+212. The Cy3 (green) and Cy5 (red) signals were obtained from Cy3 (top) and Cy5 (bottom) excitations on individual complexes of readthrough at TS (**b**), termination at TS with subsequent Cy5 PIFE (**c**), and termination at TS without Cy5 PIFE (**d**). Yellow vertical lines indicate NTP injection timing. PIFE delay (*t*_delay_) in (**c**) is the time difference between Cy3 vanishing and Cy5 PIFE starting. (**e**) PIFE occurrences plotted against TS-downstream lengths from the experiments with 150 mM NaCl. The occurrence data (black squares) fit to a 1D diffusion model (red line). (**f**) PIFE delays plotted against TS-downstream lengths from the 150 mM NaCl experiments. The delay data (black squares) fit to a 1D diffusion model (red line). (**g**) The 1D diffusion coefficients estimated using the PIFE occurrence data (red) and PIFE delay data (green) are plotted against NaCl concentrations.

**Figure 3 ijms-22-02398-f003:**
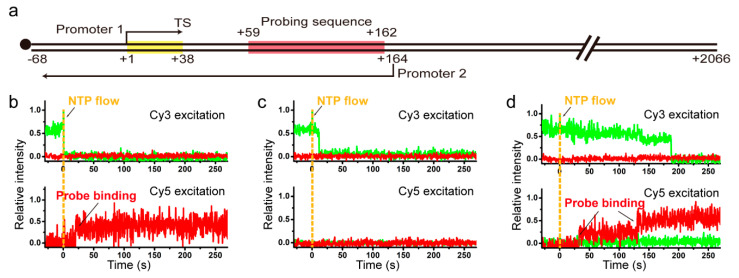
(**a**) DNA template L+2P containing two oppositely oriented promoters and a probing sequence between them. When the probing sequence is transcribed from promoter 1 or 2, transcripts can be probed with Cy5-labeled sense or antisense probe, respectively. (**b**–**d**) Representative Figure 2. P with the sense probe at Cy3 (top) and Cy5 (bottom) excitations for reinitiation at promoter 1 (**b**), for no reinitiation occurring after TS-termination (**c**), and for TS-readthrough (**d**).

## Data Availability

The data that support the findings of this study are available from the corresponding author upon reasonable request.

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
