# Peer review of "Hopping and Flipping of RNA Polymerase on DNA during Recycling for Reinitiation after Intrinsic Termination in Bacterial Transcription"

_ijms, 2021, doi:10.3390/ijms22052398_

Round 1

Reviewer 1 Report

The work of Kang and colleagues address the RNA polymerase recycling in the context of bacterial transcription. The aim of the study was to conclude about with one of the two diffusion processes hopping or flipping is mainly used for the RNAPs recycling. The methodology used is based on single molecule fluorescence approach and the use of different salt concentrations. The obtained result reinforce the fact that the RNAps recycling involve hopping diffusion process.

The manuscript is clear and the experiments are well explained. 

However, i have some comments and questions:

Figure 2 C : The presented picture corresponds to the use go a L+212 DNA, and the t delay obtained is about 25 seconds. But in the figure 2F, the PIFE delay observed is very different for a 212 TS downstream length. Even for a 512 pb, the PIFE t delay is less than 20 seconds. Why is there such a difference?

Figure 2C: Why do we observe a modification of the Cy5 signal at 120 seconds? With a relative intensity near zero? In the experiments the CY5 labelling is on DNA and not on the RNAP, so how do you explain this observation? The bleaching profil (figure S2), can't explain that. This phenomenon is not observed in the read through cases where PIFE occurs too.

Figure 2G: In think it would be useful to show the impact of salt concentration on PIFE delay data on the L+212 DNA. 

The results presented in the figure 3 are quite difficult to analyse.  Is it possible to show the results for the antigens probes?  The use of a scheme specifying the sens and antisens RNA product for the two promoters and the sens and antisens probes will help the figure gain in clarity for the understanding and analysis of your results. 

I have a naive question concerning the figure 3: Is it possible that the results presented in figure 3B are also a read through with RNA dissociation? Meaning that there is two possible read through with (3B) or without (3C) RNA releasing?

From the results presented figure 3 you conclude that the distance doesn’t really influence the recycling and that its mainly the hopping mechanism that is implicated in the reinitiation of transcription. You should try the effect of salt on L+2P template and on a longer one, to prove this. 

Syntaxe errors:

Errors: line 116: it s written immobilizaion instead of immobilization

Reviewer 2 Report

In this paper by Kang et al., the authors study the diffusion of RNAP on DNA upon reinitiation of transcription using single-molecule techniques. It is a technically sound paper that is an incremental increase compared to a recent publication of the same group (ref. 17). 

In general, it would be good to more clearly delineate the differences between these data and the ones that are published in Nat Comm 2020 and highlight the importance of the new information.

Comments:

Fig. S1: more repeats are required to get better histograms and fits. In general, the number of repeats should appear clearly in the figure legends.

Please add a schematics of the TIRF microscope in supplementary info.

l.365: why use an alternating laser system and not concomitant excitation?

Minor comments:

In general, add schematics to the figures to explain the experiments better.  

l.74: expand on "two observations of 1D diffusion of recycling RNAp apparently support different mechanisms of diffusion"

l.111: "anywhere OF DNA"

l.117: "Cy3/5 vanishing OF RNA"?

Fig.3: Define "Imager binding"

l.213-216: why is the error range broader for the delay time compared to occurence?

l.316: explain "the protons are subject to regulation"?

l.350: change to "custom-purified Cy5-labelled"

l.352: define "piranha solution"?

l.380: please redefine all terms including PR or PL to make it easier for the reader.

Round 2

Reviewer 1 Report

Hopping and flipping of RNA polymerase on DNA during re- cycling for reinitiation after intrinsic termination in bacterial transcription
 Wooyoung Kang1, Seungha Hwang2, Jin Young Kang2, Changwon Kang3* and Sungchul Hohng1* 

Round 2

Figure 2 C: The presented picture corresponds to the use of L+212 DNA, and the tdelay obtained is about 25 seconds. But in the figure 2F, the PIFE delay observed is very different for 212 bp of TS-downstream length. Even for a 512 bp, the PIFE tdelay is less than 20 seconds. Why is there such a difference? 

Following the suggestion, we replace the traces with more representative ones. 

You have not answer the question. How is it possible to observe  so different curves?

There is still some “grey areas” in the figure 3 results interpretation. You conclude (lines 277 to 286) that the prom2 reinitiation efficiency is about 24%. But your results from antisense probes shows only 10% of reinitiation at promoter 2 since there is only 0,009 of type 1 experiment with Cy5 signal and 0,901 of type 2 experiment without Cy5 signal.  In the material and methods section, part 4.3. The reasoning is based on the fact that the termination efficiency is the same in the experiments presented in  figures 2 and 3. But the techniques used are not the same and I’m not convinced. 

More over, the reinitiation at promoter 2 can also occur after the TS read-through that is observed in 40% of your sense probed experiments. Your reasoning and your conclusions of 24% of reinitiation efficiency at promoter 2 are made on a reinitiation occurring at TS site only (lanes 277 to 286). 

In order to conclude that the recycling efficiencies are similar for sense and antisense promoters and that the distance doesn’t matter, it would be usefully to try new DNA probes with a TS at equal distance between the two promoters and to show the results obtained.

There is mistakes in the axis legend of figures S6f and S6g